# A Novel Technique to Accurately Measure Mouth Opening Using 3D Electromagnetic Articulography

**DOI:** 10.3390/bioengineering9100577

**Published:** 2022-10-19

**Authors:** Franco Marinelli, Maria Florencia Lezcano, Josefa Alarcón, Pablo Navarro, Ramón Fuentes

**Affiliations:** 1Research Centre in Dental Sciences (CICO-UFRO), Dental School—Facultad de Odontología, Universidad de La Frontera, Temuco 4780000, Chile; 2Laboratorio de Cibernética, Departamento de Bioingeniería, Facultad de Ingeniería, Universidad Nacional de Entre Ríos, Oro Verde 3100, Argentina; 3Doctoral Program in Morphological Sciences, Faculty of Medicine, Universidad de La Frontera, Temuco 4780000, Chile; 4Universidad Autónoma de Chile, Santiago 4780000, Chile; 5Department of Integral Adults Dentistry, Dental School—Facultad de Odontología, Universidad de La Frontera, Temuco 4780000, Chile

**Keywords:** mouth opening, electromagnetic articulography, vector calculus ** **

## Abstract

The mouth opening is an important indication of the functionality of the temporomandibular joint (TMJ). Mouth opening is usually evaluated by asking the patient to open their mouth as wide as possible and measuring the distance between the edges of the frontal incisors with a ruler or caliper. With the advancement of technology, new techniques have been proposed to record mandibular movement. The aim of this work is to present a novel technique based on 3D electromagnetic articulography and data postprocessing to analyze the mouth opening considering distances, trajectories, and angles. A maxilla-mandible phantom was used to simulate the mouth opening movement and fixed position mouth opening. This was recorded using the AG501 3D EMA (Carstens Medizinelektronik GmbH, Bovenden, Germany). The collected data was processed using Matlab (Mathworks, Natick, MA, USA). Fix and mobile mouth opening of 1, 2, 3 and 4 cm were simulated. It was possible to evaluate the mandibular opening through the vertical distance, the Euclidean distance, the trajectory, and the opening angle. All these values were calculated and the results were consistent with expectations. The trajectory was the highest value obtained while the vertical distance was the lowest. The angle increased as the mouth opening increased. This new technique opens up new possibilities in future research since oral opening can be analyzed using multiple variables without the need to use different devices or depending on the researcher’s experience. This will make it possible to establish which parameter presents significant differences between groups of patients or between patients who have undergone some treatment.

## 1. Introduction

The mouth opening is an important indication of the functionality of the temporomandibular joint (TMJ) [1]. A limitation in the degree of opening can be symptom of TMJ dysfunction [2]. It is usually defined as the distance between the incisor crests when the mouth is open to its maximum, or as the interincisal distance plus the overbite [3]. The mouth opening is usually measured simply by asking the patient to their mouth as wide as possible and measuring the distance between the edges of the frontal incisors with a ruler [4] or caliper [5], and in some cases, the overlapping of the incisors is added [6,7]. All these methods have in common that the measurements are usually taken manually and are dependent on the experience of the professional taking them. The mouth opening continues to be measured rudimentarily, and there is still no universal criterion to include the overbite or not. 

New technologies have been proposed to perform three-dimensional analyses of the mandibular movements and thus achieve a better characterization of the behavior of the masticatory system. These offer a new possibility to analyze the mouth opening [8]. The ArcusDigma^TM^ device uses ultrasound to record mandibular movements. It has sensors mounted on a facebow that matches the mandible [9]. Ferrairo et al. [10] used a system of cameras and infrared markers to record mandibular movements. This consisted of 6 cameras and markers located on the patient’s face while the movement of the extraoral antenna was recorded. Furtado et al. [11] used a similar system, but it only had 3 cameras, and Tian et al. [12] used a two-camera system, compensating for involuntary movements using neural networks. Santos et al. [13] proposed the use of electromagnetic sensors mounted on a facebow and the use of neural networks to determine the sensor coordinates. Fuentes et al. [14] used the AG501 electromagnetic articulograph (EMA) (Carstens Medizinelektronik GmbH, Bovenden, Germany), which has a principle similar to that of Santos.

Electromagnetic articulography is an appropriate alternative for the three-dimensional analysis of mandibular movement. The AG501 provides the three-dimensional coordinates in a temporal and spatial resolution suitable for mandibular study [15], and the mounting of the sensors allows the patient to move naturally [16].

The aim of this work is to present a novel technique based on 3D electromagnetic articulography and data postprocessing to analyze the mouth opening precisely, considering distances, trajectories, and angles. 

## 2. Materials and Methods

A maxilla-mandible phantom with no metallic pieces was used to simulate the mouth opening movement. This will be recorded using the AG501 3D EMA (Carstens Medizinelektronik GmbH, Bovenden, Germany) (Figure 1). The equipment has a spatial resolution of 0.3mm and a sampling rate up to 1250 Hertz [15]. The AG501 EMA consists of 16 sensors that are small coils that can bond to the point of interest. These are subjected to a magnetic field, which induces current in them. Depending on the intensity of the induced current, the equipment determines the position within the measurement area.

Nine sensors were used: three reference, three active sensors to take measurements, and three sensors in an accessory of the articulograph called the Bite Plane. The sensors were calibrated before taking the records. 

The active sensors were placed on the interincisal midline of the frontal incisors of the mandible and on the line that divides the left and right first molar and the second premolar on the vestibular side (Figure 2). The reference sensors are typically located on the glabella and the right and left mastoids [17]. In this case, they are placed on the upper part of the phantom that represents the maxilla (Figure 2). The reference sensors eliminate from the record the movements of the head which are not of interest. The Bite Plane is an accessory used so the occlusal and horizontal planes of the coordinate system coincide. Three sensors, two lateral and one central, are placed in grooves it contains made specifically for this purpose (Figure 3(1)). 

Before taking the records of interest, a record was created where the data were collected from the reference and Bite Plane sensors used for the Head Correction procedure. Head Correction is a function of the equipment that uses these data both to eliminate the movement of the head in each recording through the reference sensors, and to list the horizontal plane of the system at the patient’s occlusal plane using the Bite Plane sensors. This can be seen in Figure 3(2,3). In the former, the reference and interincisal sensors are seen without having used the Head Correction, all below the coordinate origin. In the latter, the interincisal sensor is barely below the coordinate origin, since now the horizontal plane of the system coincides with the occlusal plane and the reference sensors are above the origin. The blue axis is the Z axis, the red one is the X axis and the green one is the Y axis. Once the Head Correction had taken place, the Bite Plane was removed, and the records were created.

### 2.1. Records

Nine records were created to be able to show different ways to assess the mouth opening at different levels. The first record was made with the phantom closed to determine the zero opening. This would be equivalent to the maximum intercuspation position (MIP). Then, 4 records were created, leading the phantom to 1, 2, 3, and 4 cm of opening approximately in the negative Z direction compared to the MIP determined in the previous point. A certain mouth opening was simulated and then a 5-second record was created (Figure 4). A second series of 4 records was created to be able to register the entire opening trajectory. The record with the closed phantom was begun and then the phantom was opened slowly until reaching the desired point: 1, 2, 3, or 4 cm approximately in the negative Z direction compared to that obtained in the first record (MIP). Once this point was reached, the recording was stopped. 

### 2.2. Data Processing

The records were stored in binary files. With the collected data, the opening was calculated in the following ways using Matlab (Mathworks, Natick, MA, USA).

#### 2.2.1. Position

The coordinates of the MIP record and the 4 fixed-position records were obtained as the average of the entire record.
(1)$$x_{j}=\overset{N}{\underset{i=1}{\sum }}\frac{x_{i}}{N}$$
(2)$$y_{j}=\overset{N}{\underset{i=1}{\sum }}\frac{y_{i}}{N}\;z_{j}=\overset{N}{\underset{i=1}{\sum }}\frac{z_{i}}{N}$$
where *x_i_*, *y_i_* and *z_i_* are the coordinates recorded by the interincisal sensor in MIP and in the opening position corresponding to 1, 2, 3 and 4 cm. *x_j_*, *y_j_* and *z_j_* is the average of each coordinate during the recording. These will define the point corresponding to each record.

#### 2.2.2. Vertical Range

The difference in coordinate *Z* was calculated between the MIP position of the interincisal sensor and the position reached for 1, 2, 3 and 4 cm.
(3)$$\;dz_{j}=z_{j}-z_{PMI}$$
*dz_j_* is the vertical distance between the position of the sensor in MIP and the sensor position in the opening corresponding to 1, 2, 3 and 4 cm.

#### 2.2.3. Euclidean Distance

The Euclidean distance between the position of the sensor in MIP and the position reached for 1, 2, 3 and 4 cm was calculated using the following equation.
(4)$$d_{j}=\sqrt{{\left(x_{j}-x_{PMI}\right)}^{2}+{\left(y_{j}-y_{PMI}\right)}^{2}+{\left(z_{j}-z_{PMI}\right)}^{2}}$$
*dj* is the Euclidean distance between the point corresponding to the opening at 1, 2, 3 and 4 cm and MIP. 

#### 2.2.4. Angle

The angle was obtained using the premolar sensors. With the interincisal sensor as a reference, two vectors were defined between this point and the premolar sensors, a perpendicular vector is obtained between them and then the angle between this vector is calculated, when the phantom is in MIP, and the same vector in the positions of interest.

The vectors are defined as follows:(5)$$\bar{V_{R}}=\bar{S_{2}S_{1}}=\langle x_{2}-x_{1},y_{2}-y_{1},z_{2}-z_{1}\rangle$$
(6)$$\bar{V_{L}}=\bar{S_{3}S_{1}}=\langle x_{3}-x_{1},y_{3}-y_{1},z_{3}-z_{1}\rangle$$
(7)$$\bar{V_{P}}=\bar{V_{L}}\times \bar{V_{R}}$$

$\bar{V_{R}}$ is the vector formed between points *S*_1_ and *S*_2_, pointing towards the right intermolar sensor, $\bar{V_{L}}$ is the vector formed between points *S*_1_ and *S*_3_, pointing towards the left intermolar sensor, and $\bar{V_{P}}$ is the vector perpendicular to both obtained using the cross product (Equation (7)). Figure 5 provides a diagram of the vector system created and angle obtained.

Using the point product between the perpendicular vector during MIP and the perpendicular vector in the positions of interest, $\bar{V_{Pi}}$, we can obtain the angle *θ*.
(8)$$\theta_{j}=\left|\cos^{-1}\left(\frac{\bar{V_{PMI}}\cdot \bar{V_{Pj}}}{\left|\bar{V_{PMI}}\right|\left|\bar{V_{Pj}}\right|}\right)\right|$$
*θ**_j_* is the angle of the openings of 1, 2, 3 and 4 cm.

#### 2.2.5. Trajectory

The point-to-point distance was calculated, and they were added to obtain the total route.
(9)$$T=\overset{N-1}{\underset{i=1}{\sum }}\sqrt{{\left(x_{i+1}-x_{i}\right)}^{2}+{\left(y_{i+1}-y_{i}\right)}^{2}+{\left(z_{i+1}-z_{i}\right)}^{2}}$$

The data generated by the articulograph were in binary files and processed in MATLAB^®^ (MathWorks Inc., Natick, MA, USA).

## 3. Results

For the MIP point, the coordinates −0.7, −1.3, and −7.7 were obtained in x, y, and z, respectively. Table 1 provides the parameters obtained on the basis of the data from each record. In Figure 6 the parameters dz and d can be seen graphically in the sagittal plane. Figure 7 illustrates the plane formed by the three sensors and how it inclines as the mandible opens.

With the second series of records, the trajectory (T) made by the mandible of the phantom was calculated and the outline of the interincisal sensor was visualized (Figure 8).

## 4. Discussion

This new technology assesses the mouth opening more accurately. Figure 9 contains different measurements used in the analysis of the mouth opening (Figure 9(1)–(4) and the relation among them (9.5). Generally, the overbite (Figure 9(1)) and the distance between the interincisal edges (Figure 9(2)) are added to estimate distance C (Figure 9(3)), resulting in an over-estimation of distance C. Creating the record with the EMA, distance D (Figure 9(4)) is in fact equivalent to target distance C.

An interesting aspect when evaluating the mouth opening is that some subjects open wider than others. In their work, Zawawi et al. [1] sought the relation between the width of the fingers on one hand and the maximum mouth opening; they reported that different studies have found a maximum mouth opening range between 40 and 60 mm. This is to be expected since people with greater body dimensions will have a larger mandible and a wider vertical opening. Al Hammad et al. [5] analyzed the correlation between the maximum mandibular opening, weight, and height, finding a positive but weak relation (r = 0.34 and r = 0.2, respectively). When comparing men and women, they found significant differences in the mouth opening, being greater in men. Park et al. [18] conducted a similar study in children, obtaining similar results. Singh et al. [19] studied the mandibular opening in 756 adults in Yamunanagar City, Haryana, India, and found that the men had a greater mandibular opening (45.36 ± 6.70 mm) than the women (41.27 ± 6.75 mm) with a significant *p*-value (*p* = 0.000). Making comparisons between subjects of different contextures can be difficult since what the maximum opening is for one subject might not be so for another. 

Several studies have analyzed the correlation between mandibular length (distance between the mandibular condyle and the incisal edge of the lower front incisor) and the mouth opening. Ingervall [20,21] found a weak correlation, r = 0.21 [20] and r = 0.3 [21]. Dijkstra et al. [22] found a correlation, r = 0.36, and Westling & Helkimo [23] obtained a correlation of r = 0.61. These differences may be due to the different ways to measure mandibular length. Ingervall took the measurements on x-ray profiles, whereas Dijkstra used a facebow and a square. For their part, Westling & Helkimo took anthropometric measurements using a ruler. Thus, the opening of each subject cannot depend on the temporomandibular joint being healthy or presenting some anomaly, but rather it is simply a question of the possible opening for the dimensions of the subject being studied. To eradicate the influence of the mandibular length or that of height, weight, or other anthropometric characteristics, the oral opening angle has been recorded to analyze the state of the temporomandibular joint. Widmalm & Larsson [24] used light sensors in their study, one of which was shared with the mandible using a mouthbow and the other fixed to the skull. Westling & Helkimo [23] and Pullinger et al. [25] calculated the opening angle from the mandibular distance using the law of cosines for an isosceles triangle. This supposes that the temporomandibular joint is fixed, which does not correspond with reality, since the condyle undergoes translation and rotation during opening [26]. Muto et al. [26] calculated the opening angle based on x-ray profiles. Dijkstra et al. [7] developed a mandibular goniometer and combined with a facebow they obtained the mandibular angle. The facebow is used to compensate for the head tilting. With this technique, Gökçe et al. [27] compared the mouth opening angle between dentate and edentulous patients, finding a significant difference (t = 5.424, *p* = 0.000) between the two. When comparing men and women, they found no significant differences (t = −0.170, *p* = 0.866). This is consistent with the observations by Westling & Helkimo [23] and Pullinger et al. [25]. However, Muto et al. [26] and Dijkstra et al. [28] found significant differences with a greater opening angle in men.

Another topic of interest in the study of the mandibular opening is the relation between this and the masticatory muscles, mainly the masseter and temporalis muscles [29]. For example, Manns et al. [30,31] used the vertical dimension [31] and the distance between the edges of the canines [30] in their study of electromyographic activity, which would be equivalent to the vertical and Euclidean distances seen here (dz and d, respectively). In addition to the electromyographic activity, the relation between the mouth opening and the bite force have also been studied. Garret et al. [32] studied the activity of the temporalis muscle under different conditions of bite force and mouth opening, measured as the interincisal distance. Su et al. [33] analyzed the bite force in relation to the maximum opening, taking this as the vertical distance between the incisal edges of the maxillary and mandibular central incisors. As with EMG [34] and force [35], to make comparisons between subjects, these parameters can be standardized by data processing. 

As indicated previously, the mouth opening is an indicator that serves to assess the state of the TMJ. The mouth opening forms part of the ranges of mandibular movement along with left and right lateral movements and contact protrusion [36]. These parameters are usually used to evaluate the mobility and state of the TMJ. It has even been suggested that there is a relation between the mouth opening and the rest of the ranges, although the results are not conclusive [37]. With electromagnetic articulography, kinematic analysis of the mandibular movement is possible, rather than static parameters. Thus, the aforementioned parameters can be recorded, and the trajectory described by the mandible from the start to the end point of the movement can also be observed [38,39]. This can open new perspectives on the diagnosis of the state of the TMJ since the trajectory and the form of the path of the mandible when performing the opening movement can be incorporated, both to analyze the maximum opening and the openings performed while chewing. Lezcano et al. [16] conducted a study on the symmetry of mandibular movements where not only the ranges of movements but also the trajectory of the mandible during mastication and ‘Posselt’s envelope of motion’ were analyzed. 

As the mouth opening is an indicator of TMJ condition, it can be use like a parameter to assess the effects of clinical procedures. For example, surgical removal is one of the most common outpatient procedure in maxillofacial surgery [40]. An inflammatory process follows these surgeries with the consequent mouth opening limitation. Thus, the extraction of the third molar in clinical trials is a study model commonly used to test the efficacy of analgesics and anti-inflammatories [41]. Several studies used mouth opening to assess the efficacy of anti-inflammatory drugs. Balakrishnan et al. [40], Paiva-Oliveira et al. [41] Momesso et al. [42] and Gursoytraket al. [43] find an important reduction in mouth opening on immediately postoperative period. However, there is less limitation of mouth opening in groups that used corticosteroid, especially in the first 24 hours after surgery [41,42,43]. The reduction of mouth opening resolves within 7–10 days after surgical procedure with ad-ministration of antibiotics and analgesics [41]. Only Paiva-Olivera and Balkrishnan report the method of mouth opening measurement, using a digital caliper to record the distance between the frontal incisors.

In previous review, we have seen that there is no consensus about the measurement of mouth opening although is widely considered. Several studies use the definition of distance between edges of front incisors not including overbite [5,19,20,40,41], or including overbite [22,23]. Others consider the vertical distance between edges of frontal incisors [18], vertical dimension [31] or distances between edges of canines [30]. The use of mouth opening angle like an alternative to assess mouth opening has led to developing measurement instruments [7,24,27,28] or techniques to estimate it [25,28]. In present study, we have demonstrated that measurement of all these parameters is possible with EMA.

The measurements of distance are usually done by caliper [5], millimetric ruler [18] or devices made for this purpose [31]. For angle measurement, the analysis is limited to one dimension [24] or static analysis [7,27] and requires large mechanism attached to subject’s head [7,24] modifying the natural movement of jaw. The use of cosines law to calculate the opening angle requires do measurement over cephalography profiles and assumes the jaw like a perfect joint [26]. In case of distance measurement, EMA can unify the analysis. Distance can be calculated like vertical or Euclidean distance using the data of the same record session. Parallax errors are digitally corrected, and measurements do not depend on human subjectivity. The size of sensors allows anatomical points to be used like a reference for the measurement and they can be placed inside the mouth to avoid skin movement during mouth opening. For the angle measurement, EMA does not require large devices attached to the subject´s head, allowing natural movement, ionizing radiation is not used and any head tilt is corrected by the Head Correction procedure. Compared to previous techniques, EMA allows a holistic analysis since it includes distance, angle, trajectory, and 3D movement and permits the standardization of the measurement procedure. This characteristic can improve the analysis of clinical trials where mandibular movement is an indicator of treatment effects [44] or drug efficacy [40,41]. Using the Bite Plane device, the EMA 501 makes it possible to define a coordinate system where the horizontal plane is aligned with the occlusal plane. This is important because it eliminates parallax errors and misalignments. A limitation of the use of the EMA 501 is that the presence of metallic elements in the dentition causes alterations in the magnetic fields, thus erroneous readings occur. This eliminates the possibility of measuring subjects with metal-based prostheses or dental implants. In addition, the sensors cannot be less than 12 mm between each other because they can generate interference with each other. The use of EMA and data postprocessing may be difficult for the clinical professional. The user must do postprocessing data. The sensors must be calibrated previously to use and when any of the sensors are changed. The costs of EMA 501 and the sensors limits the use of this equipment for investigation purposes.

Lezcano et al. [15] analyzed the accuracy and reliability of EMA 501 and compared EMA with a millimetric ruler. A support for sensors and a maxilla-mandibular phantom were used. Through Bland–Altman analysis, they found the limits agreement between 0.5 and −0.9 mm. This means that the use of EMA or a millimetric ruler is equivalent.

The Electromagnetic Articulograph has been certified by Federal Communications Commission (independent US government agency) as a low-power communication device transmitter that uses electromagnetic fields with a frequency range of 7.5 to 13.75KHz. This range is lower than the frequency range of radio transmission devices such as cellphones (10 MHz to 300 GHz) and is considered safe to human [14].

In conclusion, the present study demonstrates that it is possible to measure oral opening in several ways by means of EMA and that these coincide with that used in research where mandibular opening is analyzed. The study was made in a phantom, which do not recreate the conditions of mouth like humidity and tilt of head. A future study in humans may reveal new aspects of this technique. 

## 5. Conclusions

Electromagnetic articulography has been shown to be ideal for recording mouth opening on investigation field. This technique unifies the recording of several ways to measure mouth opening and add the 3D analysis. EMA can realize these records without need of additional devices, which allows an easy comparison between studies and may be helpful to unify the measurement procedure. EMA shows limitations in the research that involves metal implants as metal interferes with electromagnetic fields and produces wrong measurements. 

## Figures and Tables

**Figure 1 bioengineering-09-00577-f001:**
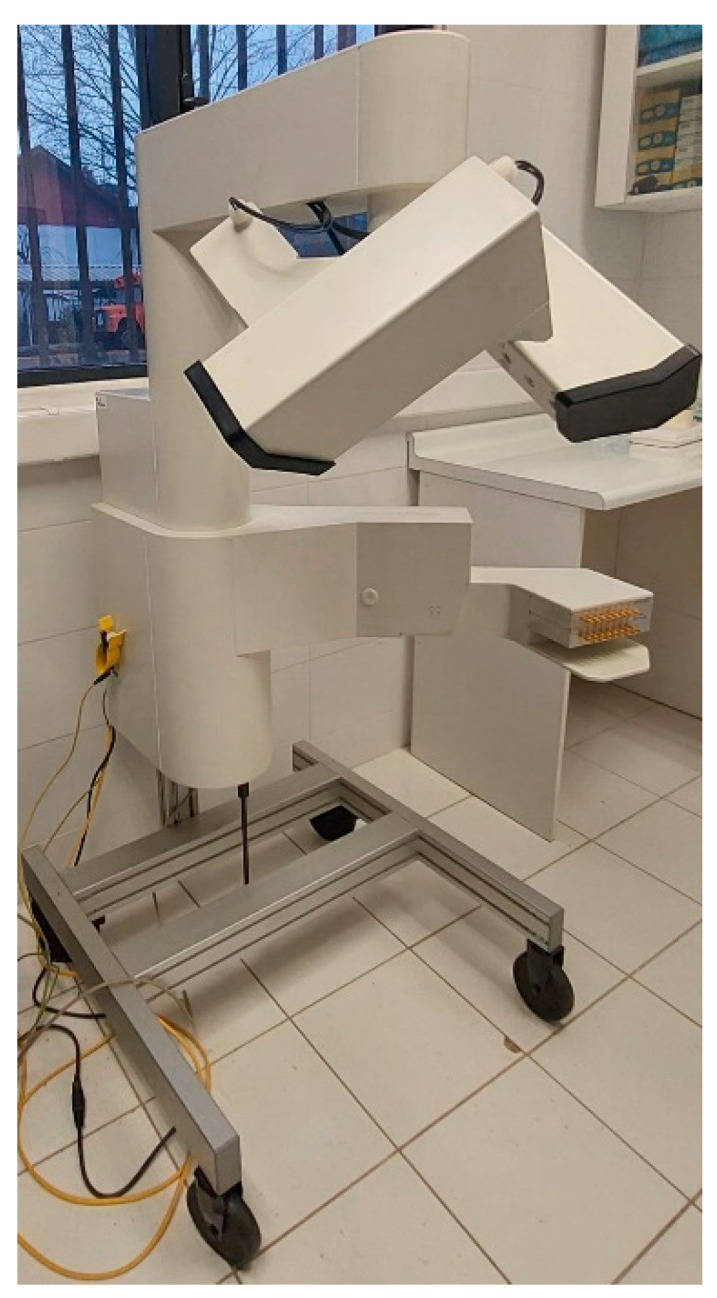
AG501 EMA.

**Figure 2 bioengineering-09-00577-f002:**
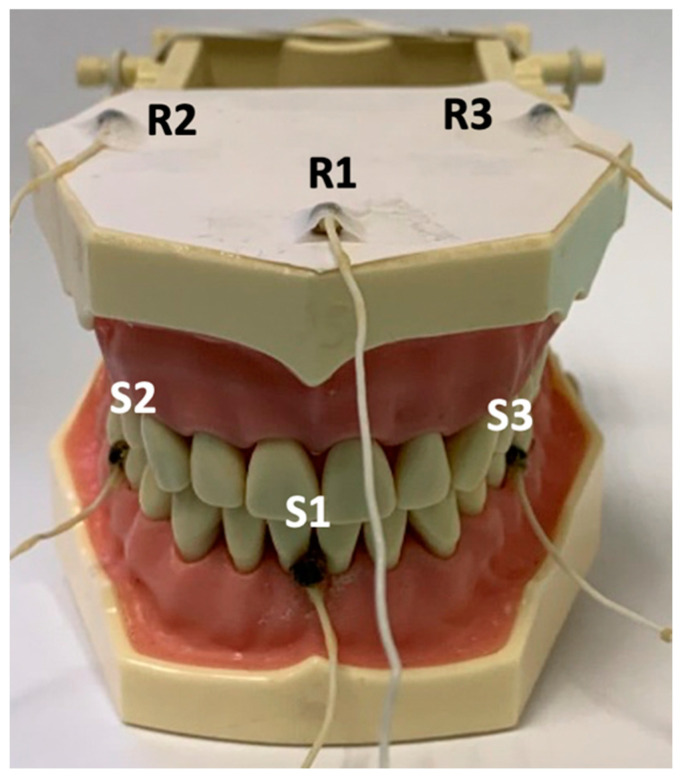
Phantom sensors. Reference sensors R1, R2, and R3. Active sensors S1, S2, and S3.

**Figure 3 bioengineering-09-00577-f003:**
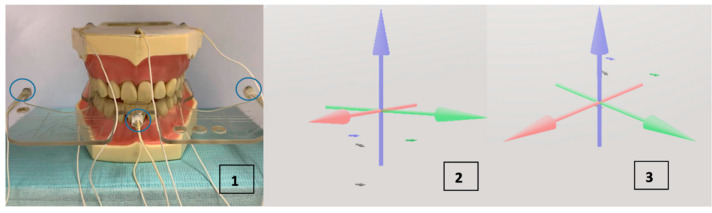
(**1**) Phantom with Bite Plane. The Bite Plane is placed on the occlusal plane and its sensors are used to transfer the origin of the system to this plane. In blue, the Bite Plane sensors. (**2**) The system without making the Head Correction. (**3**) The Head Correction done with the Bite Plane. The interincisal sensor is barely below the coordinate origin.

**Figure 4 bioengineering-09-00577-f004:**
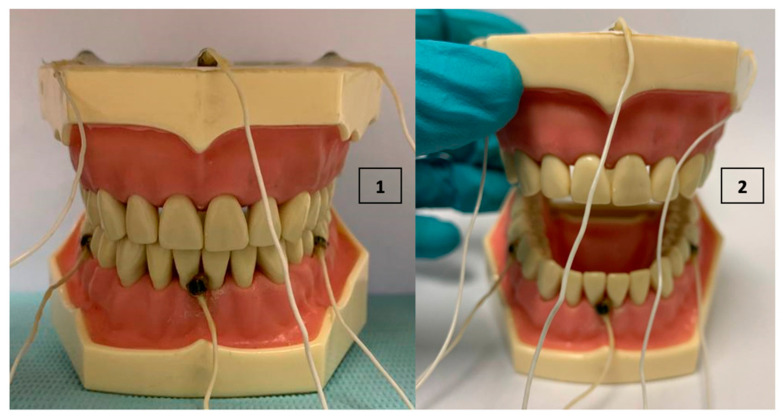
(**1**) Closed Phantom, MIP. (**2**) Manually open phantom.

**Figure 5 bioengineering-09-00577-f005:**
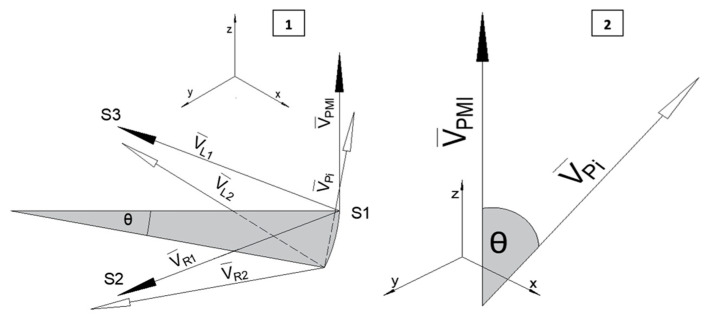
(**1**) Vectors $\bar{V_{R}}$ and $\bar{V_{L}}$ (1 in MIP, 2 in position of interest) constructed with the premolar sensors and the interincisal sensor, vector $\bar{V_{P}}$ will be perpendicular to them. The black arrows indicate when the system is in MIP, the whit arrows are with the system in the position of interest. (**2**) Diagram of the angle between the perpendicular vectors in MIP and the position of interest. The angle between them is equal to the inclination of the mandible.

**Figure 6 bioengineering-09-00577-f006:**
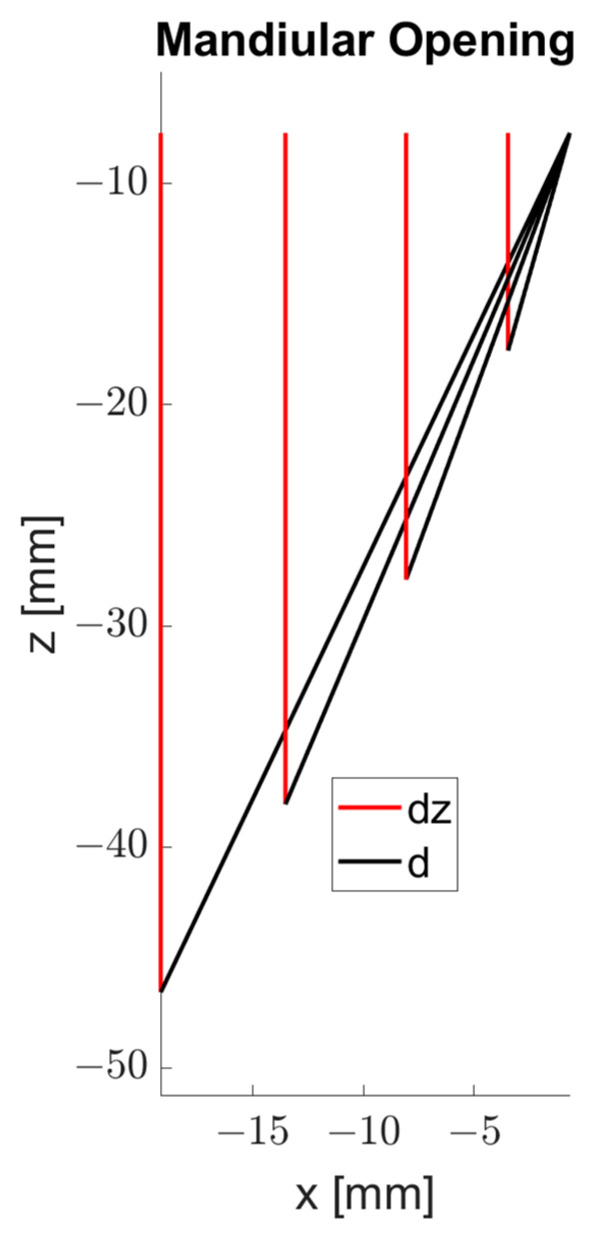
Parameters Euclidean distance (d), and vertical range (dz).

**Figure 7 bioengineering-09-00577-f007:**
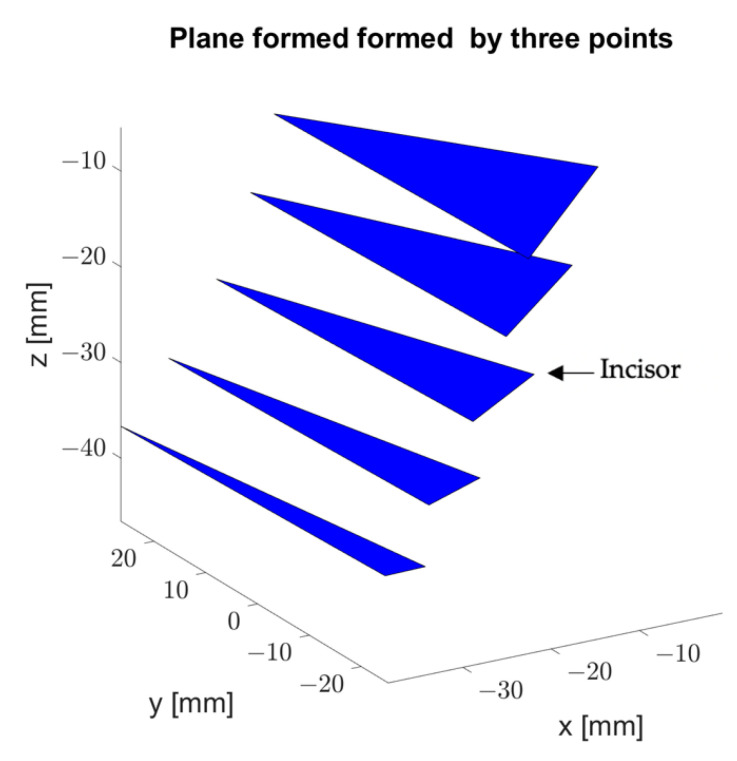
3D visualization of the plane formed by the three active sensors. The incline of the plane in each measurement is observed.

**Figure 8 bioengineering-09-00577-f008:**
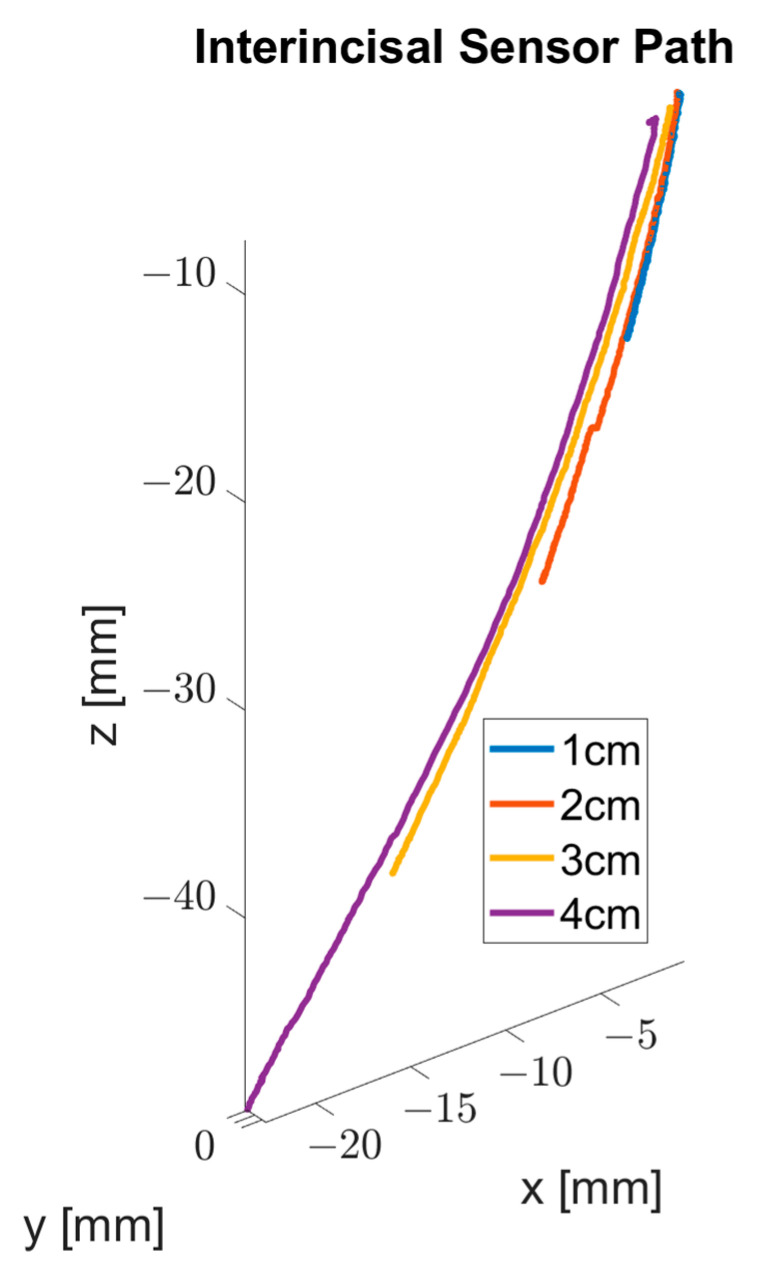
3D visualization of the outline of the interincisal sensor.

**Figure 9 bioengineering-09-00577-f009:**
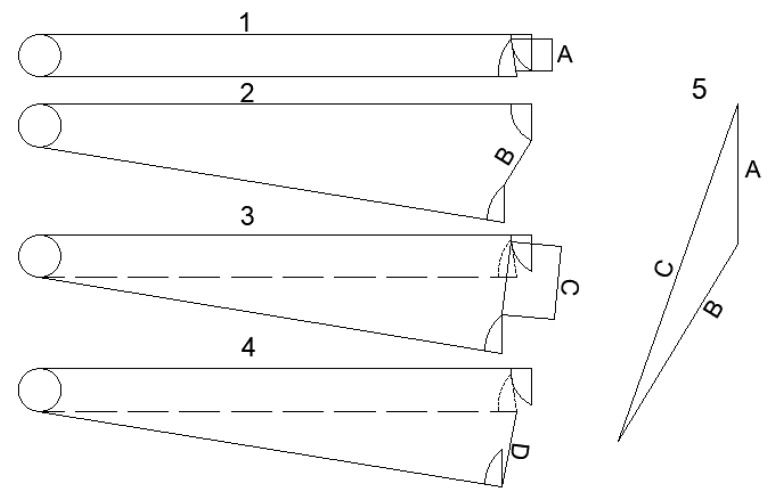
(**1**) Overbite between the upper and lower incisors, A. (**2**) Distance between the incisal edges, B. (**3**) Distance between the starting and end points of the incisal edge of the lower incisor, C. (**4**) Distance obtained from the record of the starting and end points of the interincisal sensor, D. (**5**) Diagram that shows the differences between the distance desired to be estimated, C, compared to B and A. As can be seen, C is different from both B and from the distance obtained by adding A and B.

**Table 1 bioengineering-09-00577-t001:** Parameters calculated with the collected data.

Measurement	dz [mm]	d [mm]	θ [°]	T [mm]
1 cm	9.8	10.2	5.6	16.7
2 cm	20.2	21.5	12.0	28.1
3 cm	30.4	33.0	18.3	39.0
4 cm	38.9	43.1	23.8	48.7

## Data Availability

The data presented in this study are available on request from the corresponding author. The data are not publicly available due to no public database is available.

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
