# Peer review of "A Novel Technique to Accurately Measure Mouth Opening Using 3D Electromagnetic Articulography"

_bioengineering, 2022, doi:10.3390/bioengineering9100577_

Round 1

Reviewer 1 Report

An interesting study is presented that evaluates mouth opening using a novel method. The design seems adequate and the results are clear.

It is suggested to add in the discussion comment on the advantages and disadvantages in relation to the techniques used to assess mouth opening, including cost, ease, comfort, safety, accuracy, reproducibility, etc.

Also, it is suggested to add a comment on the relevance and/or usefulness in relation to the evaluation of anti-inflammatory drugs after third molar surgery, since mouth opening is a frequently evaluated indicator of efficacy.

Author Response

REVIEWER 1

Thank you so much for reviewing our manuscript. Your suggestions have been highly helpful to us. You may find our reply (R) below each comment (C).

C1. An interesting study is presented that evaluates mouth opening using a novel method. The design seems adequate and the results are clear.

R1. Thank you, we are pleased by your comment.

C2. It is suggested to add in the discussion comment on the advantages and disadvantages in relation to the techniques used to assess mouth opening, including cost, ease, comfort, safety, accuracy, reproducibility, etc.

R2. Thank you for your suggestion. We have added the next paragraphs to discussion to consider these points.

In this paragraph are treated the topics related to advantages and comfort. The measurements of distance are usually done by caliper [1], millimetric ruler [2] or devices made for this purpose [3]. For angle measurement, the analysis is limited to one dimension [4] or static analysis [5, 6] and requires large mechanism attached to subject’s head [4, 5] modifying the natural movement of jaw. The use of cosines law to calculate the opening angle requires do measurement over cephalography profiles and assumes the jaw like a perfect joint [7]. In case of distance measurement, EMA can unify the analysis. Distance can be calculated like vertical or Euclidean distance using the data of the same record session. Parallax errors are digitally corrected and measurement not depends of human subjectivity. Size of sensors allows use anatomical points like reference measurement and they can be placed inside mouth to avoid the skin movement during mouth opening. To angle measurement, EMA does not require large devices attached to subject´s head, allowing natural movement, ionizing radiation is not used and tilt head is corrected by the Head Correction procedure.

In this paragraph are treated the topics related with accuracy and reproducibility. Lezcano et al. [8] analyzed the accuracy and reliability of EMA 501 and compared EMA with a millimetric ruler. A support for sensors and a maxilla-mandibular phantom were used. Through Bland-Altman analysis, they found the limits agreement between 0.5 and -0.9 mm. This means that the use of EMA or a millimetric ruler is equivalent.

In this paragraph are treated the topic related with security. The Electromagnetic Articulograph has been certified by Federal Communications Commission (independent US government agency) as a low-power communication de-vice transmitter that uses electromagnetic fields with a frequency range of 7.5 to 13.75KHz. This range is lower than the frequency range of radio transmission devices such as cell-phones (10 MHz to 300 GHz) and is consider safe to human [9].

In this paragraph are treated the topic related with costs and limitations are treated and the implications of these.A limitation of the use of the EMA 501 is that the presence of metallic elements in the dentition causes alterations in the magnetic fields, thus erroneous readings occur. This eliminates the possibility of measuring subjects with metal-based prostheses or dental implants. In addition, the sensors cannot be less than 12 mm between each other because they can generate interference with each other. The use of EMA and data postprocessing may be difficult for the clinical professional. The user must do postprocessing data. The sensors must been calibrated previously to be used and when any of sensors are changed. The costs of EMA 501 and the sensors limits the use of this equipment for investigation purpose.

C3. Also, it is suggested to add a comment on the relevance and/or usefulness in relation to the evaluation of anti-inflammatory drugs after third molar surgery, since mouth opening is a frequently evaluated indicator of efficacy.

R3. Thank you for your suggestion. We have added the next paragraph to include the use of mouth opening in evaluation of anti-inflamatory drugs.

As the mouth opening is an indicator of TMJ condition, it can be use like a parameter to assess the effects of clinical procedures. For example, surgical removal is one of the most commonly outpatient procedure in maxillofacial surgery [10]. An inflammatory process follows these surgeries with the consequent mouth opening limitation. Thus, the extraction of the third molar in clinical trials is a study model commonly used to test the efficacy of analgesics and anti-inflammatories [11]. Several studies used mouth opening to assess the efficacy of anti-inflammatory drugs. Balakrishnan et al. [10], Paiva-Oliveira et al. [11] Momesso et al. [12] and Gursoytraket al. [13] find an important reduction in mouth opening on immediately postoperative period. However, there is less limitation of mouth opening in groups that used corticosteroid, especially in the first 24 hours after surgery [11-13]. The reduction of mouth opening resolves within 7–10 days after surgical procedure with administration of antibiotics and analgesics [11]. Only Paiva-Olivera and Balkrishnan report the method of mouth opening measurement, using a digital caliper to record the distance between the frontal incisors.

Reviewer 2 Report

A Novel Technique to Accurately Measure Mouth Opening Using 3D Electromagnetic Articulography

The aim of this work is to present a novel technique based on 3D electromagnetic articulography and data postprocessing to analyze the mouth opening precisely, considering distances, trajectories, and angles.

Comments:

Introduction:

In the introduction, the study provides a justification for the presentation of the new technique for measure mouth opening using 3d electromagnetic articulography. The text briefly describes some studies that approach new technologies to perform three-dimensional analyzes of the mandibular movements.

Aim:

In the presentation of the study objectives, the last sentence containing aspects associated with the inference of results can be removed from this place (... Results obtained from tests with a phantom are presented and the advantages of this technique compared to other methods will be in phased.)

Materials and Methods:

The description of the methodology is detailed and presented in stages. A maxilla-mandible phantom with no metallic pieces was used to simulate the mouth opening movement. Nine sensors were used: three reference, three active sensors to take measurements, and three sensors in an accessory of the articulography called the Bite Plane. The sensors were calibrated before taking the records.

Results:

The results showed the parameters obtained based on the data from each record.

Discussion:

In the discussion, the authors may remove the first sentence as it repeats the objectives and is not required.

In this study, the novel technique was presented based on 3D electromagnetic articulation and data postprocessing to analyze the mouth opening accurately, considering distances, trajectories, and angles.

In the second, third and fourth paragraphs, the authors present a review of the literature on the topic in question. This part can be enriched with a better correlation and detailed discussion with the new technique proposal - A Novel Technique to Accurately Measure Mouth Opening Using 3D Electromagnetic Articulography (Thus, the discussion must be rewritten).

Considering a laboratory study, it is interesting to make the clinical relevance and possible limitations clear to readers.

Conclusions:

The conclusion of the study must present the clinical benefits of the technique, the possible limitations and inferences, and there is no need for any additional comments to be made here. Thus, the conclusions must be rewritten.

Author Response

Thank you so much for reviewing our manuscript. Your suggestions have been highly helpful to us. You may find our reply (R) below each comment (C).

C1. In the introduction, the study provides a justification for the presentation of the new technique for measure mouth opening using 3d electromagnetic articulography. The text briefly describes some studies that approach new technologies to perform three-dimensional analyzes of the mandibular movements.

R1. Thank you for your comment.

C2. In the presentation of the study objectives, the last sentence containing aspects associated with the inference of results can be removed from this place (... Results obtained from tests with a phantom are presented and the advantages of this technique compared to other methods will be in phased.)

R2. Thank you for your comment. We have removed the next sentence from introduction.

Results obtained from tests with a phantom are presented and the advantages of this technique compared to other methods will be emphasized.

C3. The description of the methodology is detailed and presented in stages. A maxilla-mandible phantom with no metallic pieces was used to simulate the mouth opening movement. Nine sensors were used: three reference, three active sensors to take measurements, and three sensors in an accessory of the articulography called the Bite Plane. The sensors were calibrated before taking the records.

R3. Thank you for your comment.

C4. The results showed the parameters obtained based on the data from each record.

R4. Thank you for your comment.

C5. In the discussion, the authors may remove the first sentence as it repeats the objectives and is not required.

R5. Thank you for the suggestion. We have removed the next sentence from discussion.

In this study, a novel technique was presented based on 3D electromagnetic articulography and data postprocessing to analyze the mouth opening accurately, considering distances, trajectories, and angles.

C6. In the second, third and fourth paragraphs, the authors present a review of the literature on the topic in question. This part can be enriched with a better correlation and detailed discussion with the new technique proposal - A Novel Technique to Accurately Measure Mouth Opening Using 3D Electromagnetic Articulography (Thus, the discussion must be rewritten).

R6. Thank you for the suggestion. We have added the next paragraphs to do a better comparative with the techniques seen in the paragraph 2, 3 and 4.

The measurements of distance are usually done by caliper [1], millimetric ruler [2] or devices made for this purpose [3]. For angle measurement, the analysis is limited to one dimension [4] or static analysis [5, 6] and requires large mechanism attached to subject’s face [4, 5] modifying the natural movement of jaw. The use of cosines law to calculate the opening angle requires do measurement over cephalography profiles and assumes the jaw like a perfect joint [7]. In case of linear measurement, EMA can unify the analysis. Distance can be calculated like vertical or Euclidean distance using the data of the same record session. Parallax errors are digitally corrected and measurement not depends of human subjectivity. Size of sensors allows use anatomical points like reference to measurement and they can be placed inside mouth to avoid the skin movement during mouth opening. To angle measurement, EMA does not require large devices attached to subject´s head, allowing natural movement, ionizing radiation is not used and tilt head is corrected by the Head Correction procedure.

C7. Considering a laboratory study, it is interesting to make the clinical relevance and possible limitations clear to readers.

R7. Thank you for the suggestion.

Referred to clinical relevance we added this sentence in the paragraph of R6.  Compared to previous techniques, EMA allows a holistic analysis since include distance, angle, trajectory and 3D movement and permits standardization of measurement procedure. This characteristic can improve the analysis of clinical trials where mandibular movement is an indicator of treatment effects [8] or drugs efficacy [9, 10].

Referred to limitations of EMA we added the next paragraph. A limitation of the use of the EMA 501 is that the presence of metallic elements in the dentition causes alterations in the magnetic fields, thus erroneous readings occur. This eliminates the possibility of measuring subjects with metal-based prostheses or dental implants. In addition, the sensors cannot be less than 12 mm between each other because they can generate interference with each other. The use of EMA and data postprocessing may be difficult for the clinical professional. . The user must do postprocessing data. The sensors must been calibrated previously to use and when any of sensors are changed. The costs of EMA 501 and the sensors limits the use of this equipment for investigation purpose.

C8. The conclusion of the study must present the clinical benefits of the technique, the possible limitations and inferences, and there is no need for any additional comments to be made here. Thus, the conclusions must be rewritten.

R8. Thank you for the suggestion. We have rewritten the conclusions to include only the mentioned topics.

Electromagnetic articulography has been shown to be ideal for recording mouth opening on investigation field. This technique unifies the recording of several ways to measure mouth opening and add the 3D analysis. EMA can realize these records without need of additional devices, which allows an easy comparison between studies and may be helpful to unify the measurement procedure. EMA shows limitations in the research that involves metal implants as metal interfere with electromagnetic fields and produce wrong measurements.

Reviewer 3 Report

Dear Authors, first of all I would like to congratulate You on your work. The topic Is of great clinical relevance. However, I believe that the article could be improved. Please, take a note of some suggestions.

Abstract: is precisely written, and the aim of the study is mentioned. Please include some more information about the results/finding to enhance the impact of this section.

The introduction; is detailed, compact, covering the background information and the rationale of the study effectively. However, the last paragraph is very details and suggested to condense that information.

Materials and methods- this section is well organized. 

Statistical analysis is very basic. Only correlations were calculated

Discussion- this paragraph should be rearranged. It is very chaotic. Please do not repeat information from Introduction and try to be more focused. Rewrite this section using following paragraphs: main results and clinical relevance; comparison with other studies; advantages and disadvantages of the study; conclusions and suggestions for future studies.

Conclusions- try to simplify this section.

I believe that your manuscript would have much more relevance after suggested improvements.

Author Response

Thank you so much for reviewing our manuscript. Your suggestions have been highly helpful to us. You may find our reply (R) below each comment (C).

C1. Abstract: is precisely written, and the aim of the study is mentioned. Please include some more information about the results/finding to enhance the impact of this section.

R1. Thank you for the suggestion. We have added the next sentences to abstract to show more details about results.

Fix and mobile mouth opening of 1, 2, 3 and 4 cm were simulated… All these values were calculated and the results were consistent with expectations. The trajectory was the highest value obtained while the vertical distance was the lowest. The angle increased as the mandibular opening increased.

C2. The introduction; is detailed, compact, covering the background information and the rationale of the study effectively. However, the last paragraph is very details and suggested to condense that information.

R2. Thank you for the suggestion. We have deleted the next sentences from introduction to condense the information.

As they are not directly above the mandible, the sensors do not provide the real coordinates of a point of interest, but rather they record the movement of the sensor, which matches that of the mandible.

…which is held to the vestibular side of the mandibular incisors.

… but which enables the sensors to adhere directly to the point of interest and to correct movements automatically.

C3. Materials and methods- this section is well organized.

R3. Thank you for the comment.

C4. Statistical analysis is very basic. Only correlations were calculated.

R4. Thank you for the comment. The present study is a proof of context (POC). A POC is an exercise in which work is focused on determining whether an idea can be turned into reality. In this work we demonstrated that is possible measure mouth opening with EMA and data postprocessing, thus statistical analysis was not necessary. The accuracy of EMA was evaluated and founded suitable by Lezcano et al. [18].

C5. Discussion- this paragraph should be rearranged. It is very chaotic. Please do not repeat information from Introduction and try to be more focused. Rewrite this section using following paragraphs: main results and clinical relevance; comparison with other studies; advantages and disadvantages of the study; conclusions and suggestions for future studies.

R5. Thank you for the comment. To no repeat information from introduction and when compare with other techniques, we have deleted the next sentences.

Paragraph 1. In this study, a novel technique was presented based on 3D electromagnetic articu-lography and data postprocessing to analyze the mouth opening accurately, considering distances, trajectories, and angles.

Paragraph 3. The technique based on electromagnetic articulography presented in this work seeks a simpler way to measure the mandibular angle and allows the patient more natural movements, in addition to eliminating the use of ionizing radiation.

Paragraph 3. With the AG501 it is possible to evaluate both parameters, mouth opening and mouth opening angle, using the data from one opening record. In addition, the translation of the mandible and the rotation on the three axes is recorded, obtaining a complete record of the movements of the mandible, both translation and rotation.

Paragraph 4. Using electromagnetic articulography it is possible to conduct similar studies, for both force analysis and electromyographic activity, and, using data from the same record, ana-lyze the mandibular opening as the vertical distance, the Euclidean distance or the open-ing angle without needing to resort to other measuring instruments.

We added the next paragraph to resume the main result of the study

In previous review, we have seen that there is no consensus about the measurement of mouth opening although is widely consider.  Several studies use the definition of distance between edges of front incisors not including overbite [1-5], or including overbite [6, 7]. Others consider the vertical distance between edges of frontal incisors [8], vertical dimension [9] or distances between edges of canines [10]. The use of mouth opening angle like an alternative to assess mouth opening has led to develop measurement instruments [11-14] or techniques to estimate it [14, 15]. In present study, we have demonstrated that measurement of all these parameters are possible with EMA.

We added the next paragraph to do a comparative with techniques seen in the review

The measurements of distance are usually done by caliper [1], millimetric ruler [8] or devices made for this purpose [9]. For angle measurement, the analysis is limited to one dimension [12] or static analysis [11, 13] and requires large mechanism attached to subject’s face [11, 12] modifying the natural movement of jaw. The use of cosines law to calculate the opening angle requires do measurement over cephalography profiles and assumes the jaw like a perfect joint [16]. In case of linear measurement, EMA can unify the analysis. Distance can be calculated like vertical or Euclidean distance using the data of the same record session. Parallax errors are digitally corrected and measurement not depends of human subjectivity. Size of sensors allows use anatomical points like reference to measurement and they can be placed inside mouth to avoid the skin movement during mouth opening. To angle measurement, EMA does not require large devices attached to subject´s head, allowing natural movement, ionizing radiation is not used and tilt head is corrected by the Head Correction procedure.

We added the next paragraph to analyze the clinical relevance of using EMA to assess mouth opening

Compared to previous techniques, EMA allows a holistic analysis since include distance, angle, trajectory and 3D movement and permits standardization of measurement procedure. This characteristic can improve the analysis of clinical trials where mandibular movement is an indicator of treatment effects [17] or drugs efficacy [4, 5].

We added the next paragraph to trait the conclusions, disadvantages of the study and considerations for future studies.

In conclusion, the present study demonstrates that it is possible to measure oral opening in several ways by means of EMA and that these coincide with that used in research where mandibular opening is analyzed. The study was made in a phantom, which do not recreate the conditions of mouth like humidity and tilt of head. A future study in humans may reveal new aspects of this technique.

C6. Conclusions- try to simplify this section.

R6. Thank you for the comment. We have rewritten the conclusions.

Electromagnetic articulography has been shown to be ideal for recording mouth opening on investigation field. This technique unifies the recording of several ways to measure mouth opening and add the 3D analysis. EMA can realize these records without need of additional devices, which allows an easy comparison between studies and may be helpful to unify the measurement procedure. EMA shows limitations in the research that involves metal implants as metal interfere with electromagnetic fields and produce wrong measurements.